Validity and reliability of inertial sensors for elbow and wrist range of motion assessment

Costa Vanina 1 2 vanina.costacortez@ceu.es
Ramírez Óscar 2
http://orcid.org/0000-0003-4568-2933 Otero Abraham 1
http://orcid.org/0000-0001-7367-2551 Muñoz-García Daniel 3
Uribarri Sandra 4
Raya Rafael 1 2
1 Escuela Politécnica Superior, Universidad San Pablo-CEU, CEU Universities , Madrid , Spain
2 Werium Assistive Solutions Ltd. , Madrid , Spain
3 Motion in Brains Research Group, Neuroscience and Motion Science Institute, The Center for Advanced Studies University La Salle (Universidad Autónoma de Madrid) , Madrid , Spain
4 The Center for Advanced Studies University La Salle, Universidad Autónoma de Madrid , Madrid , Spain
Daumer Martin
Electronic publication date: 2020 Aug 11
Publication date: 2020
Volume: 8
Electronic Location ID: e9687
Received 2020 Feb 26; Accepted 2020 Jul 18
Copyright: © 2020 Costa et al.
Copyright year: 2020
Copyright holder: Costa et al.
License: This is an open access article distributed under the terms of the Creative Commons Attribution License, which permits unrestricted use, distribution, reproduction and adaptation in any medium and for any purpose provided that it is properly attributed. For attribution, the original author(s), title, publication source (PeerJ) and either DOI or URL of the article must be cited.
License URL: https://creativecommons.org/licenses/by/4.0/

Keywords: Inertial sensors, Range of motion, Joint assessment, Goniometer, Reliability, Elbow joint, Wrist joint

Funding: Ministry of Science, Innovation and Universities of Spain RTI2018-095324-B-I00, RTI2018-097122-A-I00 and IDI-20191120 European Regional Development Fund of the European Commission This research was funded by the Ministry of Science, Innovation, and Universities of Spain grant numbers RTI2018-095324-B-I00, RTI2018-097122-A-I00 and IDI-20191120 and by the European Regional Development Fund of the European Commission. The funders had no role in study design, data collection and analysis, decision to publish, or preparation of the manuscript.

==============================
Background

Elbow and wrist chronic conditions are very common among musculoskeletal problems. These painful conditions affect muscle function, which ultimately leads to a decrease in the joint’s Range Of Motion (ROM). Due to their portability and ease of use, goniometers are still the most widespread tool for measuring ROM. Inertial sensors are emerging as a digital, low-cost and accurate alternative. However, whereas inertial sensors are commonly used in research studies, due to the lack of information about their validity and reliability, they are not widely used in the clinical practice. The goal of this study is to assess the validity and intra-inter-rater reliability of inertial sensors for measuring active ROM of the elbow and wrist.

Materials and Methods

Measures were taken simultaneously with inertial sensors (Werium™ system) and a universal goniometer. The process involved two physiotherapists (“rater A” and “rater B”) and an engineer responsible for the technical issues. Twenty-nine asymptomatic subjects were assessed individually in two sessions separated by 48 h. The procedure was repeated by rater A followed by rater B with random order. Three repetitions of each active movement (elbow flexion, pronation, and supination; and wrist flexion, extension, radial deviation and ulnar deviation) were executed starting from the neutral position until the ROM end-feel; that is, until ROM reached its maximum due to be stopped by the anatomy. The coefficient of determination (r2) and the Intraclass Correlation Coefficient (ICC) were calculated to assess the intra-rater and inter-rater reliability. The Standard Error of the Measurement and the Minimum Detectable Change and a Bland–Altman plots were also calculated.

Results

Similar ROM values when measured with both instruments were obtained for the elbow (maximum difference of 3° for all the movements) and wrist (maximum difference of 1° for all the movements). These values were within the normal range when compared to literature studies. The concurrent validity analysis for all the movements yielded ICC values ≥0.78 for the elbow and ≥0.95 for the wrist. Concerning reliability, the ICC values denoted a high reliability of inertial sensors for all the different movements. In the case of the elbow, intra-rater and inter-rater reliability ICC values range from 0.83 to 0.96 and from 0.94 to 0.97, respectively. Intra-rater analysis of the wrist yielded ICC values between 0.81 and 0.93, while the ICC values for the inter-rater analysis range from 0.93 to 0.99.

Conclusions

Inertial sensors are a valid and reliable tool for measuring elbow and wrist active ROM. Particularly noteworthy is their high inter-rater reliability, often questioned in measurement tools. The lowest reliability is observed in elbow prono-supination, probably due to skin artifacts. Based on these results and their advantages, inertial sensors can be considered a valid assessment tool for wrist and elbow ROM.

Introduction

Human motion capture has many applications in fields such as the automotive industry, 3D animation, sports, activities of daily living and physiotherapy (Ahmad et al., 2013). In the medical field, human movement is mainly important for the diagnosis of neuro-motor disorders and follow-up on physical training or therapy interventions. Musculoskeletal disorders are a diversified group of conditions that affect the normal function of the skeletal system. They comprise an extensive range of conditions; some of them are acute and short-lived, while others result in progressive deterioration and disability. Between 20% and 33% of the world’s population suffers from some painful musculoskeletal condition (GBD 2017 Disease and Injury Incidence and Prevalence Collaborators, 2018). Joints involved suffer from pain, diminished ROM and physical impairment. This situation leads to mobility and functional limitations, preventing the subject from performing daily activities normally.

Although kinematic parameters such as acceleration or velocity can play a role in evaluating the patient’s motor control or functional assessment, the most common parameter used in the clinical practice to characterize the human joint is the range of motion (ROM). The purpose of this paper is to study the validity of inertial sensors to measure the elbow and wrist active ROM. Elbow and wrist chronic conditions are very common among musculoskeletal problems. However, the measure of the kinematic of the upper limb can be complex due to the multiple degrees of freedom of their joints.

Both for upper and lower limb, the goniometer is the simplest and most extended tool for measuring ROM in the clinical practice (Vauclair et al., 2018). It was developed approximately 60 years ago, and its versatility and usability led to this instrument being promptly integrated into the field of physiotherapy and rehabilitation as a helpful evaluation tool (Gajdosik & Bohannon, 1987; Roach et al., 2013). However, some authors question the poor intertester reliability of the universal goniometer. Chapleau et al. (2011) have assessed its validity compared with radiographic measurements. They measured ROM in flexion, extension and carrying angle of the elbow using three repetitions for each movement. The intraclass correlation coefficients (ICC) ranged from 0.94 to 0.97 from the goniometric measurements and from 0.98 to 0.99 for the radiographic measurements. Despite the high ICC, the authors reported maximal errors of 10° when the ROM was measured using the goniometer (10.3° for extension, 7.0° for flexion and 6.5° for carrying angle). Clinicians also report drawbacks such as the need for holding manually the arms of the goniometer while taking a measurement, the stabilization of these during the readout, and the experience required to locate landmarks in the patient’s body (Roach et al., 2013; Mehta et al., 2017).

The evolution of the universal goniometer is its digital version, often called the inclinometer. Kolber et al. (2012) studied the reliability of the goniometer and the digital inclinometer to measure shoulder flexion, abduction, internal and external rotation. Two physiotherapists participated in the study. Results showed an excellent ICC for goniometry (>0.94) and digital inclinometer (>0.95) between the two examiners.

Technological advances in medical instrumentation have enabled more precise and specific motion capture systems that provide a more complete and objective characterization of the patient’s mobility. Photogrammetric and inertial systems are the most advanced technologies in the field of biomechanics (Yahya et al., 2019). Besides the ROM, they can provide useful information such as velocity, acceleration and other derived kinematic and dynamic parameters. Photogrammetric systems, also called optical-tracking systems (Vicon™, Qualysis™, OptiTrack™, etc.), are considered the gold standard for ROM measurement. However, they have drawbacks such as high economical and computational cost, marker occlusion, the requirement of an instrumented environment, and time-consuming setups, which prevent their usage in clinical practice (Tsushima, Morris & McGinley, 2003; Mehta et al., 2017).

The evolution of microsensors has given rise to a new miniatured generation of Inertial Measurement Units (IMUs) based on micro-electro-mechanical systems. These devices can be successfully used for accurate, non-invasive and portable motion tracking (Filippeschi et al., 2017). IMUs or inertial sensors are traditionally comprised of accelerometers and gyroscopes. An accelerometer contains a mass suspended by a spring and placed in some housing. Changes in the mass displacement caused by movement are related directly with changes in acceleration. A gyroscope measures the angular velocity by using vibration to determine the orientation. When a magnetometer is integrated, they are also named magneto-inertial sensors.

Inertial sensors-based systems are becoming very precise. Additionally, their relative affordability respect to optical-tracking systems, the faster and simpler setups, and the advantage of not being confined to a laboratory setting, make them ideal tools for assessing ROM in the clinical practice (Ahmad et al., 2013). There are several reviews which explore the usage of these instruments for a variety of health purposes, yielding interesting research in biomechanics, such as the classification of age-related kinematic impairment (Greene et al., 2015; Roldán-Jiménez & Cuesta-Vargas, 2016), body mobility assessment (Iwasaki & Hirotomi, 2015), gait analysis and development of athletes’ performance (Camomilla et al., 2018) and sports (Boddy et al., 2019; Wells et al., 2019).

In the context of neuromotor disorders, its usefulness has been proved even in the most serious motor dysfunctions, such as is the case of neurological motor disorders (Iwasaki & Hirotomi, 2015; Raya et al., 2015), paraplegic limbs (Wiesener et al., 2019) or osteoarthritis (Chen et al., 2015). Cerebral palsy, stroke, traumatic brain injury, and motor neuron disease are some of the conditions where the use of inertial sensors in rehabilitation therapies has proven to be beneficial (Bai et al., 2015; Bertomeu-Motos et al., 2015). They also play a promising role in quantifying patients’ ROM and daily living activities such as reaching and grasping (Ertzgaard et al., 2016). There are also promising results testing their validity and reliability to assess the ROM of the cervical joint in subjects with CP (Carmona-Pérez et al., 2020).

Concerning the upper limb biomechanics, there are several studies testing the validity of inertial sensors. Robert-Lachaine et al. (2017) conducted a validation of an inertial system (Xsens™) compared to an optoelectronic system (OptoTrak™). They studied the whole-body kinematics from 12 participants performing some functional movements (elbow flexion/extension and pronation/supination and wrist flexion/extension, lateral deviation and circumduction). Although in general they observed acceptable results, they reported the largest errors for the elbow and shoulder, and they attributed them to the different biomechanical models more than to the technological precision. They concluded that caution should be taken for comparison involving IMUs and optoelectronic systems.

Ertzgaard et al. (2016) studied a set of coordination tasks comparing an inertial sensing system with an optical tracking system. Results showed small systematic errors in all the studied movement planes ranging between 1.2° and 1.3°. The lowest intraclass correlations coefficients were obtained for elbow prono-supination (Ertzgaard et al., 2016). Sacco et al. (2015) used an inertial sensing system to measure the upper limb ROM in 77 elderly patients. In this case, the authors concluded that the system seemed less reliable and valuable compared to the traditional methods used in gerontology, but not significant differences were found between planes of movement (Sacco et al., 2015).

There are a few studies analyzing the validity of inertial sensing systems of elbow and wrist ROM for each plane of movement, but most of these studies have usually been carried out under lab conditions (Muller et al., 2017; Robert-Lachaine et al., 2017). Some of them are preliminary studies without a statistical sample size (Wells et al., 2015). Others do not conduct a study of validity and reliability of the inertial sensors’ measurements with the goniometer’s (Tian et al., 2015; Zhou et al., 2008; Zhang, Wong & Wu, 2011), something essential to test whether it is possible to replace the goniometer with this new technology. There is also a lack of standardized protocols for motion analysis other than for gait because most of the studies were conducted under lab conditions. The transition from a research instrument to a clinical reliable tool is one of the main challenges concerning inertial sensing systems and the evaluation of upper limb function (Ertzgaard et al., 2016).

There also are studies that focus on the usage of smartphones, that integrate inertial sensors, for ROM measurement. They are an easy-to-use and accessible tool, making them practical in clinical assessments. While some studies describe a high reliability between universal goniometer and smartphones assessment, others reflect a large heterogeneity when analyzing each movement separately. Behnoush et al. (2016) obtained a good validity measure for a smartphone compared with a goniometer when used to measure elbow supination (ICC = 0.92), but more disappointing results were obtained when used to measure elbow flexion (ICC = 0.73). This study addressed only the validity of the measures, and not their inter-rater and intra-rater reliability. Furthermore, the measurements carried out with the goniometer were not performed by trained physiotherapists who routinely use this device at work, but by doctors who received training on how to use the goniometer to carry out the experiment. In another similar study, Vauclair et al. (2018) stated that the smartphone app (Clinometer™) overestimated flexion (mean 6.4° +− 1.0°) and supination (5.9° +− 1.9°) respect to the universal goniometer. However, there was not a significant difference in pronation.

The goal of this article is to validate inertial sensors for elbow and wrist active ROM measurement. We shall analyze their validity when compared with the measurements taken by physiotherapists using a goniometer, as well as their intra-rater and inter-rater reliability. For that purpose, the Werium™ motion capture system was employed. It is based on inertial technology; more specifically, on the ENLAZA™ inertial sensor (Raya et al., 2012). This device has already proven to be an accurate solution for cervical ROM measurement (Raya et al., 2018). However, the validity and reliability of this system on other joints, specifically elbow and wrist, had not been previously studied.

Materials and Methods

Study design

This study was designed to assess criterion-related validity and intra-inter-rater reliability of ROM measures for the wrist and elbow. Criterion-related validity is related to the capability of one instrument to deliver results as close as possible to the gold standard to which it is being compared. Reliability aims to measure consistency between repeated measurements of the same variable on the same individual, who is subjected to the same conditions. While intra-rater reliability deals with the consistency of one individual evaluating the same fact on two different occasions, inter-rater reliability focuses on measuring the variability of two or more raters evaluating the same fact on the same occasion.

The clinical validation was performed in the Center for Advanced Studies University La Salle (IRF La Salle, Madrid, Spain), which granted ethical approval (cseuls-pi-146/2017). Written consent was obtained from the patients to carry out the study. Two physiotherapists (raters “A” and “B”) were responsible for interacting with the patient and taking readings from the goniometer, as well as placing the inertial sensors on the patient’s body. Additionally, a technician was responsible for taking measurements from the inertial sensors by means of a personal computer, as well as typing in a template both the measurements from the inertial sensors and the measurements provided by the physiotherapists from the goniometer. All the subjects were individually evaluated in two successive sessions with an interval of 48 h. They performed the distinctive movements of each joint in an active way. Both sensors and goniometer were placed simultaneously; in this way, it was guaranteed that both devices were measuring the same movement. This process was repeated by rater A followed by rater B in random order. The complete description of the study, which is explained below, was performed identically for both elbow and wrist.

Participants

Participants were recruited by the IRF La Salle. Inclusion criteria included men and women between the age of 20 and 40 with approximately normal body mass index (<25 kg/m2), and without any joint pain. Exclusion criteria were pregnancy, history of joint pain either during the last 8 months or during data collection, and noticeable deformity of the target joint. A total of 29 asymptomatic adults participated in the study. A total of 20 of them were men, with an average age of 24.10 ± 3.86, height of 1.79 ± 0.08 m, weight of 76.19 ± 12.71 kg, and body mass index of 23.67 ± 3.19 kg/m2. The remaining 9 were women with an average age of 21.33 ± 1.50, height of 1.62 ± 0.09 m, weight of 58.78 ± 9.65 kg, and body mass index of 22.16 ± 1.77 kg/m2. Table 1 presents the subjects’ demographic information.

Table 1 Subject’s demographic characteristics.

Subjects	n	Age (years)	Height (m)	Weight (kg)	BMI (kg/m2)	
Male	20 (69%)	24.10 ± 3.86	1.79 ± 0.08	76.19 ± 12.71	23.67 ± 3.19	
Female	9 (31%)	21.33 ± 1.50	1.62 ± 0.09	58.78 ± 9.65	22.16 ± 1.77	
Notes:

BMI, body mass index.

Values presented as mean ± standard deviation.

Data collection instruments

Measures were taken by the Winkelmesser™ plastic goniometer manufactured by Kirchner and Wilhelm GmbH & Co. with 1° increments, and by the Werium™ motion capture system (Fig. 1). This system consists of two inertial sensors; one must be placed in the distal part of the extremity (moving sensor) and the other in the proximal part (fixed sensor). These sensors have an accuracy of ±1° (Raya et al., 2018). The angle measured will be the relative angle between both sensors. By placing a sensor in the proximal part of the extremity the possible compensatory movements of the patient are ignored when performing the measurement with the inertial sensors.

Figure 1 Werium™ inertial sensor.

The software Pro Motion Capture™ was used to take the ROM measures and to visualize these data in real-time (Fig. 2). The measurements of both inertial sensors are sent via Bluetooth to the PC where the data acquisition software is running, and from the measurements of both sensors, the software calculates the ROM of the joint. A specific template was created to support the process of data gathering. The random order in which raters had to assess participants in each session was generated by the statistical software GraphPad.

Figure 2 Screen of the software Pro Motion Capture™ during a wrist asessment.

Virtual human models display real-time movements while equivalent ROM values are simultaneously reflected in (A) sagittal, (B) transverse and (C) frontal planes.

Procedure

Before start taking measurements, the subject’s sex, age, height and weight were registered. Then, the subject was asked about his/her dominant arm, for elbow and wrist evaluation. The two raters inspected the joint and agreed on the anatomic landmarks in which the goniometer and sensors should be placed. The subject’s skin was marked with eyeliner when necessary to facilitate the recognition of these landmarks (Fig. 3). The goniometer was placed following bibliographic guidelines for an ordinary ROM assessment of elbow and wrist (Norkin & White, 2009).

Figure 3 Anatomical landmarks for goniometer alignment before measuring of elbow and wrist.

(A) Bony landmarks on the longitudinal axis of the humerus, the forearm (radius) and the lateral epicondyle. (B) Bony landmarks on the ulnar styloid and the lateral midline of the fifth metacarpal. (C) Bony landmarks on the capitate and the third metacarpal.

In the case of the elbow, the moving sensor was placed on the posterior part of the forearm two cm cranial from the radial and ulnar styloid. The fixed sensor was positioned at the lateral line of the arm (between the acromion and the lateral epicondyle of the elbow). Elbow assessment involved flexion and pronation-supination movements. Concerning the wrist, the moving sensor was placed in the middle of the posterior side of the hand (above the metacarpals). The fixed sensor was placed on the posterior part of the forearm two cm cranial from the radial and ulnar styloid. Flexion-extension and radial-ulnar deviation were the distinctive movements in the wrist. Both goniometer and inertial sensors were placed preserving the free active mobility of the subject, without any interference with the movements.

The examiner explained and performed the movement for the subject, familiarizing the participant with the activity. Regarding elbow flexion, the neutral position was standing upright in an anatomical position with the palms of the hands extended facing forward and the arms facing the sides of the body. During the flexion movement, the subject flexed the elbow by directing the palm of his/her hand towards the shoulder to avoid moving the elbow away from the side of the body and to always keep the palm of the hand outstretched (Fig. 4). During elbow pronation-supination, the starting position was seated in a chair, bringing the side of the body closer to a stretcher or a flat surface. The forearm rested on the stretcher and the arm was attached to the body. Arm and forearm formed 90°. The hand had to protrude from the supporting surface. The hand was placed sideways, with the thumb pointing towards the ceiling and the rest of the fingers of the hand extended together. During pronation the subject rotated his/her hand, trying to make the palm face the ground (Fig. 5). Flexion-extension wrist movements were evaluated while seated in a chair, bringing the side of the body closer to the stretcher or a flat surface. The forearm rested on the stretcher. The hand had to protrude from the supporting surface. In the neutral position, the hand was placed with the back facing the ceiling, fingers extended and together. During wrist flexion, the subject moved the hand, trying to keep the tips of the fingers pointed towards the floor and keeping the hand stiff. During wrist extension, the subject moved the hand, trying to keep the tips of the fingers pointed towards the ceiling. In radial-ulnar deviation, the starting position was the same, but the forearm and palm of the hand were supported on the stretcher. During the radial deviation, the subject made a lateral tilt, towards the direction in which his/her thumb was, moving his/her fingers away from the middle line that crosses the hand longitudinally. During the ulnar deviation, the subject made a lateral tilt, towards the direction in which his/her little finger was, moving the fingers away from the midline (Fig. 6).

Figure 4 Measurement of elbow flexion ROM using the goniometer and inertial sensors.

(A) Starting position for flexion-extension ROM assessment. (B) Measuring of maximum flexion ROM.

Figure 5 Measurement of elbow prono-supination ROM using the goniometer and inertial sensors.

(A) Starting position for prono-supination ROM assessment. (B) Measuring of maximum pronation ROM. (C) Measuring of maximum supination ROM.

Figure 6 Measurement of wrist ROM using the goniometer and inertial sensors.

(A) Starting position for flexion-extension ROM assessment. (B) Measuring of maximum flexion ROM. (C) Measuring of maximum extension ROM. (D) Starting position for ulnar-deviation ROM assessment. (E) Measuring of maximum radial deviation ROM. (F) Measuring of maximum ulnar deviation ROM.

There was not any warm-up exercise, but the subject was encouraged to repeat the movement with the examiner to make sure the task was understood and there was no cause of pain. All the movements were active and, consequently, the examiner did not help the participant to attain his/her ROM end-feel. For both joints, each at its own time, the technician indicated which rater should perform the evaluation in the first position (according to the random order previously generated), while the other waited outside the room. The selected rater placed the goniometer, as well as the inertial sensors, in the subject’s joint. Once all these steps had been taken, the subject was positioned in the initial or neutral position and the technician was asked to calibrate the sensors in such a way that the device considered this point as zero degrees. At this stage, the subject started the movement actively. The rater, who was also ready with the goniometer in a zero position, followed the movement of the subject’s limb with the moving arm of the goniometer and encouraged the subject to make the maximum effort in order to achieve the end of the active range. When this point was reached, the examiner read the measurement of the goniometer and communicated it to the technician, who simultaneously read, in the software tool, the measure registered by the sensors. Both measurements were collected in the template and the subject was instructed to return to the zero position. This procedure was repeated three times calibrating between repetitions. After these repetitions, the procedure followed with the next movement planned to complete the set of movements. All the data collected is available as Supplemental Materials of this article (Datas S1 and S2).

Statistical analysis

The statistical analysis was performed using IBM® SPSS Statistics statistical package, version 25. A Shapiro–Wilk test was carried out to test the normality of the data distributions. This test is recommended for a sample size of less than 50 (Ahmad & Khan Sherwani, 2015). The statistical significance level was set at p < 0.05. Mean and standard deviation (SD) were computed subsequently to characterize the ROM measured with the goniometer and the inertial sensors for elbow and wrist. For that purpose, the average value of three repetitions of each distinctive movement was used.

To evaluate validity and reliability, the correlation coefficients r2 and ICC were calculated (Portney & Watkins, 2015). ICC model (3, k) for the intra-rater analysis and the ICC model (2, k) for the inter-rater analysis were used. The Standard Error of the Measurement (SEM) and the Minimum Detectable Change (MDC) at the 90% confidence level were calculated for both measuring instruments (McKenna, Cunningham & Straker, 2004). SEM can be estimated from the ICC: SEM=SD1−ICC,

where SD is the SD of the measures. MDC is useful to characterize the effectiveness of a device to detect changes in the measured variable ant it can be calculated from the SEM (Portney & Watkins, 2015): MDC90=1,65×SEM×2.

Finally, a Bland–Altman analysis was conducted. This graphical method analyzes the agreement between two different measurement methods. It uses the mean and the SD to create a scatterplot with limits of agreement, in which the difference between the two paired measurements is plotted against the mean of the two measurements. While the mean difference is the estimated bias, the SD measures the random fluctuations around this mean (Doğan, 2018).

Results

The Shapiro–Wilk test indicated that the distributions with which we are working were in all cases normal (p-value < 0.05). Table 2 shows the mean and the SD for the measures obtained with the goniometer and the sensor for all the movements of both joints. Pronation and supination ROM for the elbow had the greatest variation.

Table 2 Descriptive statistics of elbow and wrist ROM measured by goniometer and inertial sensor, organized by rater.

		Elbow mean ROM (°) ± SD (°)	Wrist mean ROM (°) ± SD (°)	
		Flexion	Pronation	Supination	Flexion	Extension	Radial deviation	Ulnar deviation	
A	Goniometer	150° ± 7°	64° ± 11°	67° ± 14°	82° ± 10°	68° ± 9°	27° ± 7°	43° ± 8°	
Sensor	150° ± 8°	61° ± 10°	66° ± 13°	83° ± 11°	67° ± 9°	27° ± 7°	43° ± 8°	
B	Goniometer	150° ± 9°	63° ± 11°	69° ± 13°	82° ± 11°	66° ± 9°	26° ± 7°	43° ± 8°	
Sensor	150° ± 9°	61° ± 10°	67° ± 13°	83° ± 11°	66° ± 9°	27° ± 6°	43° ± 9°	
Notes:

ROM, range of motion.

SD, standard deviation.

Tables 3 and 4 present r2 and ICC, with its 95% confidence interval (CI), between the measures taken simultaneously by the goniometer and the inertial sensors. Regarding the elbow results, r2 values ranged between 0.47 and 0.95 involving all the movements in the two sessions. The lowest values for r2 were found in pronation (0.47) during the first session for examiner B, although examiner A obtained higher values. ICC values varied from 0.78 to 0.99. In the case of the wrist, r2 and ICC values varied between 0.82–0.96 and 0.95–0.99, respectively.

Table 3 r2 and ICC (with 95% CI) for the elbow ROM measurements taken by the goniometer and the sensors organized by rater and session.

	First session	Second session	
	r2	ICC	CI 95%	r2	ICC	CI 95%	
Rater A							
Flexion	0.68	0.90	[0.78–0.95]	0.72	0.90	[0.79–0.95]	
Pronation	0.59	0.84	[0.62–0.93]	0.76	0.91	[0.76–0.96]	
Supination	0.81	0.94	[0.86–0.97]	0.95	0.99	[0.97–0.99]	
Rater B							
Flexion	0.64	0.89	[0.77–0.95]	0.83	0.95	[0.90–0.98]	
Pronation	0.47	0.78	[0.51–0.90]	0.70	0.91	[0.81–0.96]	
Supination	0.90	0.97	[0.93–0.99]	0.81	0.94	[0.87–0.97]	

Table 4 r2 and ICC (with 95% CI) for the wrist ROM measurements taken by the goniometer and the sensors organized by rater and session.

	Firsi session	Second session	
	r2	ICC	CI 95%	r2	ICC	CI 95%	
Rater A							
Flexion	0.94	0.98	[0.96–0.99]	0.94	0.99	[0.97–0.99]	
Extension	0.95	0.99	[0.97–0.99]	0.91	0.98	[0.95–0.99]	
Radial deviation	0.87	0.97	[0.93–0.99]	0.89	0.97	[0.93–0.93]	
Ulnar deviation	0.88	0.97	[0.93–0.98]	0.96	0.99	[0.97–0.99]	
Rater B							
Flexion	0.92	0.98	[0.95–0.99]	0.94	0.98	[0.95–0.99]	
Extension	0.89	0.97	[0.93–0.98]	0.85	0.96	[0.91–0.98]	
Radial deviation	0.82	0.95	[0.89–0.98]	0.92	0.98	[0.95–0.99]	
Ulnar deviation	0.91	0.97	[0.94–0.99]	0.92	0.98	[0.95–0.99]	

The intra-rater and inter-rater reliability results of the elbow are shown in Tables 5 and 6. The intra-rater analysis of the elbow assessed by the goniometer yielded ICC values ranging from 0.62 to 0.91, while the sensors’ values ranged from 0.79 to 0.96. In both cases the lowest values belong to pronation movements. With regard to inter-rater reliability of elbow ROM, ICC values for the sensors ranged between 0.94 and 0.97, while for the goniometer ranged between 0.83 and 0.95.

Table 5 Intra-rater reliability analysis of goniometer and inertial sensor assessing elbow ROM.

	Goniometer	Sensor	
	ICC	CI 95%	SEM	MDC90	ICC	CI 95%	SEM	MDC90	
Rater A									
Flexion	0.86	[0.71–0.94]	4°	8°	0.79	[0.56–0.90]	5°	13°	
Pronation	0.74	[0.44–0.88]	8°	19°	0.86	[0.71–0.93]	5°	13°	
Supination	0.96	[0.90–0.98]	4°	10°	0.92	[0.84–0.97]	5°	12°	
Rater B									
Flexion	0.75	[0.46–0.88]	6°	14°	0.83	[0.63–0.92]	5°	12°	
Pronation	0.62	[0.21–0.82]	10°	23°	0.89	[0.77–0.95]	5°	11°	
Supination	0.91	[0.80–0.95]	6°	14°	0.96	[0.90–0.98]	4°	9°	

Table 6 Inter-rater reliability analysis of goniometer and inertial sensor assessing elbow ROM.

	Goniometer	Sensor	
	ICC	CI 95%	SEM	MDC90	ICC	CI 95%	SEM	MDC90	
First session									
Flexion	0.86	[0.71–0.93]	4°	9°	0.95	[0.89–0.97]	2°	6°	
Pronation	0.93	[0.85–0.97]	5°	11°	0.95	[0.89–0.94]	3°	8°	
Supination	0.95	[0.90–0.98]	4°	10°	0.96	[0.91–0.98]	4°	9°	
Second session									
Flexion	0.92	[0.82–0.96]	3°	8°	0.97	[0.93–0.98]	3°	6°	
Pronation	0.83	[0.54–0.93]	6°	14°	0.94	[0.88–0.97]	4°	8°	
Supination	0.91	[0.74–0.96]	6°	14°	0.97	[0.94–0.99]	3°	7°	

Regarding the intra-rater analysis of the wrist (Table 7), ICC values were between 0.81 and 0.93 both for the goniometer and the inertial sensors. The inter-rater analysis of the wrist yielded slightly greater ICC values for the inertial sensors (0.93–0.99) compared with those from the goniometer (0.92–0.97) (Table 8).

Table 7 Intra-rater reliability analysis of goniometer and inertial sensor assessing wrist ROM.

	Goniometer	Sensor	
	ICC	CI 95%	SEM	MDC90	ICC	CI 95%	SEM	MDC90	
Rater A									
Flexion	0.93	[0.85–0.97]	4°	9°	0.93	[0.86–0.97]	4°	9°	
Extension	0.84	[0.65–0.92]	5°	12°	0.82	[0.62–0.92]	5°	12°	
Radial deviation	0.81	[0.60–0.91]	4°	10°	0.83	[0.63–0.92]	4°	9°	
Ulnar deviation	0.85	[0.67–0.93]	4°	10°	0.87	[0.72–0.94]	4°	10°	
Rater B									
Flexion	0.93	[0.85–0.97]	4°	10°	0.91	[0.80–0.96]	5°	11°	
Extension	0.89	[0.76–0.95]	4°	10°	0.89	[0.76–0.95]	4°	10°	
Radial deviation	0.88	[0.75–0.94]	3°	8°	0.88	[0.74–0.94]	3°	7°	
Ulnar deviation	0.81	[0.58–0.91]	5°	12°	0.81	[0.59–0.91]	5°	12°	

Table 8 Inter-rater reliability analysis of goniometer and inertial sensor assessing wrist ROM.

	Goniometer	Sensor	
	ICC	CI 95%	SEM	MDC90	ICC	CI 95%	SEM	MDC90	
First session									
Flexion	0.95	[0.89–0.98]	3°	8°	0.97	[0.94–0.99]	2°	6°	
Extension	0.92	[0.80–0.96]	4°	9°	0.93	[0.86–0.97]	3°	7°	
Radial deviation	0.92	[0.83–0.96]	2°	6°	0.96	[0.92–0.98]	2°	4°	
Ulnar deviation	0.92	[0.84–0.96]	3°	7°	0.95	[0.90–0.98]	3°	6°	
Second session									
Flexion	0.97	[0.93–0.98]	3°	7°	0.99	[0.98–0.99]	1°	3°	
Extension	0.94	[0.85–0.97]	4°	8°	0.97	[0.93–0.98]	2°	5°	
Radial deviation	0.97	[0.90–0.99]	2°	4°	0.97	[0.94–0.99]	2°	4°	
Ulnar deviation	0.95	[0.90–0.98]	2°	6°	0.96	[0.91–0.98]	2°	6°	

The SEM and MDC90 for both elbow and wrist for the sensors are for most types of movements less than or equal to those of the goniometer (Tables 5–8), as it was expected since these values are vary inversely with the magnitude of the ICC, and the ICC of the sensors for most of the cases is equal to or greater than that of the goniometer.

Figures 7 and 8 show the Bland–Altman plots for the different movements of the elbow and wrist. These plots reflect a good agreement between the measurements of the inertial sensors and the goniometer.

Figure 7 Bland–Altman plots for both instruments in elbow assessment.

The figure shows the dispersion graphs comparing the goniometer and inertial sensors for (A) elbow flexion, (B) elbow supination and (C) elbow pronation. The means of both instruments are presented on the X axis, while the difference between them is presented on the Y axis. It can be noted that the majority of the values represented are distributed within the limits of agreement.

Figure 8 Bland–Altman plots for both instruments in wrist assessment.

The figure shows the dispersion graphs comparing the goniometer and inertial sensors for (A) wrist flexion, (B) wrist extension, (C) wrist radial deviation and (D) wrist ulnar deviation. The means of both instruments are presented on the X axis, while the difference between them is presented on the Y axis. It can be noted that the majority of the values represented are distributed within the limits of agreement.

Discussion

According to Table 2, similar active ROM values were obtained for the goniometer and sensors both for elbow and wrist. When using a t-test to compare the distributions of the measurements of each instrument for pronation and supination, which had the greatest variability between raters, the p-values were 0.28 for pronation and 0.56 for supination; that is, there are no statistically significant differences between the measurements of both instruments. For the rest of the movements, the mean value of the measures obtained with the sensor and the goniometer had a maximum difference of 1°, even for the measures taken by the two different physiotherapists, which is an excellent agreement.

The results of Table 2 were compared with the normal ROM values of the literature. Flexion elbow ROM, for both the goniometer and the inertial sensors, ranged between 150° ± 7° and 150° ± 9°; very similar to the 150° value reported by the American Academy of Orthopaedic Surgeons (AAOS) (Norkin & White, 2009). Pronation and supination were lower on average than those of the AAOS and the American Medical Association (AMA), which reported 80° of pronation and supination ROM. Since this affected both goniometric and sensors measurements, it was thought it could have been due to a low flexion capacity of some subjects performing such movements or due to lack of motivation during the experiment. In the case of the active ROM of the wrist, all the movements presented values consistent with the ranges of normality described in the literature, both measured by goniometer and sensors. These values were slightly higher when compared to the AAOS and AMA values, but close to those from Green and Wolf’s ROM studies (Greene & Wolf, 1989). For instance, in the case of radial deviation, the present experiment yielded values ranging from 26° ± 7° to 27° ± 7°, comparable with 25.4° ± 2° from the later, while AAOS and AMA reported 20°.

Table 3 shows a high concordance between the measurements taken by the goniometer and the sensors (ICC > 0.78 always, for most of the cases ICC > 0.90). The lowest ICC were obtained for pronation movements. Pronation involves great arthrokinematics complexity since the radius is placed above the ulna during this movement, unlike the supination where the bones are parallel. This hinders its measurement with any instrument since the instruments cannot rotate in unison with the bone to accurately measure the degrees of rotation. We also believe that the movement of the forearm’s skin could be affecting the measurements of the inertial sensors. This hypothesis seems to be supported by Muller et al. (2017), who found higher RMS errors for pronation and supination related to flexo-extension when using manual alignment of the sensors. Placing the sensor on subject’ hand instead of forearm might have improved the results of prono-supination measurement.

Intraclass Correlation Coefficient was particularly good for the wrist (ICC ≥ 0.95), considering all the movements in both sessions (Table 4). With regard to reliability, the assessment of the elbow with sensors yielded an ICC ≥ 0.83 and ICC ≥ 0.94 for intra-rater and inter-rater analysis respectively, higher than those resulting from the goniometer measurements (Tables 5 and 6). The SEM and MDC90 values for both elbow and wrist were lower in the case of the sensors than in the case of the goniometer, particularly in the inter-rater analysis of both joints (Tables 6 and 8). This suggests that the sensors can identify subtler ROM variations when compared to the goniometer.

In the literature we can find papers that use inertial sensors to estimate the movement of the upper limb (Tian et al., 2015; Zhou et al., 2008; Zhang, Wong & Wu, 2011; Robert-Lachaine et al., 2017; Ertzgaard et al., 2016), but many of them do not conduct a study of the validity and reliability of their ROM measurements compared with the measurements obtained with a goniometer. The only paper we have found with which we can directly compare our results is Behnoush et al. (2016). In this study inertial sensors of an iPhone™ were compared with a goniometer for elbow’s ROM measurement. The iPhone™ reliability values reported were greater than ours (ICC between 0.95 and 0.98); however those of the goniometer were lower than ours (ICC between 0.77 and 0.91), which also led to smaller ICC values in the assessment of concurrent validity: ICC of 0.84, 0.90 and 0.96 for flexion, pronation and supination, compared with our ICC values of 0.90, 0.88 and 0.97 (average value for both raters). The goniometer reliability values of Behnoush et al. (2016) are lower than ours likely because their measurements were carried out by doctors who received training on how to measure the ROM with the goniometer to conduct the study, while in our case physiotherapists with years of experience in the usage of the goniometer took the measurements. Behnoush’s study only analyzed intra-rater validity of the elbow, not inter-rater validity, thus no comparison with our inter-rater results is possible. We have not found any study assessing the validity and reliability of wrist ROM measurements taken with inertial sensors with which we could compare our results.

Regarding the limitations of the present study, the intra-rater outcomes could have been limited by the absence of a warm-up phase. This type of practice is commonly implemented and can improve reliability since there are specific conditions that can influence the joint ROM from one session to the next. Moreover, during the testing phase it was observed that, in some cases where the participants had great joint flexibility, the sensor was not able to capture accurately the rotation of the forearm because the skin was stretched and twisted and the sensor strap could not be held in the proper position. This can be corrected with the use of anti-slip silicone printed in the inner side of the strap, which would have resulted in greater adherence to the arm. A reduction in the size of the inertial sensors could also be an improvement. This would permit a more stable placing of the sensors, resulting in more accurate measurements.

The symmetrical movements of the joint (radial-ulnar deviation, flexion-extension, etc.) were performed partially rather than completely. That was the case for both elbow and wrist. For instance, in the case of wrist deviations, three repetitions of the radial deviation and then three repetitions of the ulnar deviation were performed, always returning to the neutral starting point. An alternative would have been to perform the full or complete range, that is, three repetitions of maximum deviation from radial to ulnar. There is evidence that this second procedure could yield more precise ROM because in the first case the subject’s limb does not exactly return to its starting point for each new repetition of the motion (Raya et al., 2018).

Conclusions

The measurements of the elbow and wrist ROM taken with inertial sensors have a high agreement with those taken with a goniometer. For the elbow the ICC values range from 0.78 to 0.99, while for the wrist they range from 0.95 to 0.99 (Tables 3 and 4). Furthermore, the ROM values obtained with the sensors are in agreement with the normal ROM values published in the literature (Table 2). These findings support the validity of their measures.

The reliability of their measurements in intra-rater scenarios compared to that of a goniometer is similar or slightly superior, with ICC values ranging from 0.62 to 0.96 for the elbow, and from 0.83 to 0.95 for the wrist in the case of the goniometer, compared with ICC values ranging from 0.83 to 0.96 for the elbow, and from 0.94 to 0.97 for the wrist in the case of the sensors (Tables 5 and 6). In inter-rater scenarios, the reliability of the sensors is higher compared to that of the goniometer for the elbow; while the ICC values of the goniometer range from 0.92 to 0.97, for the sensors they range from 0.93 to 0.99 (Table 8). In the case of the wrist the inter-rater reliability is similar in both cases, ranging from 0.81 to 0.93 (Table 7).

Overall, our results support the usage of inertial sensors to evaluate the ROM of elbow and wrist. Furthermore, inertial sensors present advantages over goniometry with regard to the easiness of usage, which in case of the goniometer is more dependent on the experience of the practitioner locating anatomical references or performing the reading of measurements (Chapleau et al., 2011). They also avoid the need for holding the arms of the goniometer while taking a measurement, or for stabilization during the end-feel ROM readout. And, given that the measurements they provide are already digital, they can be recoded automatically using a PC, and even incorporated into the patient’s electronic medical record. All these advantages make inertial sensors an interesting solution to replace the traditional goniometer.

The easy connectivity of inertial sensors with computing devices enables their usage in virtual rehabilitation scenarios through videogames and/or virtual environments. Given than rehabilitation success is greatly dependent on patient’s motivation, innovative upper-limb rehabilitation procedures combining inertial sensors with serious games are being currently addressed (Giggins, Sweeney & Caulfield, 2014; Callejas-Cuervo, Gutierrez & Hernandez, 2017). As future work we plan to explore the usage of inertial sensors and serious video games for the rehabilitation of the upper limb. We also would like to repeat this study with patients, instead of asymptomatic volunteers. This would provide a deeper understanding of specific musculoskeletal data and exercise rehabilitation programs for the elbow and wrist joints.

Supplemental Information

Supplemental Information 1 Elbow ROM raw data.

Click here for additional data file.

Supplemental Information 2 Wrist ROM raw data.

Click here for additional data file.

The authors of this work would like to thank the volunteers of the experimental trials for coming and contributing selflessly to this project.

Additional Information and Declarations

Competing Interests

Author Contributions

Human Ethics

Data Availability

Rafael Raya is the CEO of Werium Solutions; Vanina Costa is a PhD student at Werium Solutions; Óscar Ramírez works at Werium Solutions. Rafael Raya, Vanina Costa, and Óscar Ramírez are developers of Pro Motion Capture™ software.

Vanina Costa conceived and designed the experiments, performed the experiments, analyzed the data, prepared figures and/or tables, authored or reviewed drafts of the paper, and approved the final draft.

Óscar Ramírez conceived and designed the experiments, performed the experiments, prepared figures and/or tables, technical support, and approved the final draft.

Abraham Otero analyzed the data, authored or reviewed drafts of the paper, and approved the final draft.

Daniel Muñoz-García conceived and designed the experiments, authored or reviewed drafts of the paper, and approved the final draft.

Sandra Uribarri performed the experiments, prepared figures and/or tables, and approved the final draft.

Rafael Raya conceived and designed the experiments, analyzed the data, authored or reviewed drafts of the paper, and approved the final draft.

The following information was supplied relating to ethical approvals (i.e., approving body and any reference numbers):

The Center for Advanced Studies University La Salle (Universidad Autónoma de Madrid) granted Ethical approval to carry out the study within its facilities (cseuls-pi-146/2017).

The following information was supplied regarding data availability:

Data are available at Zenodo: Vanina, Costa, Óscar, Ramírez, Sandra, Uribarri, & Rafael, Raya. (2020). Elbow and wrist range of motion assessment comparing inertial sensors against goniometry_Raw data clinical validation (Data set). Zenodo. DOI 10.5281/zenodo.3683250.

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
