# Peer review of "Validity and reliability of inertial sensors for elbow and wrist range of motion assessment"

_PeerJ, doi:10.7717/peerj.9687_

## Round 0.1 · original submission · Major Revisions

You have seen that one reviewer has recommended to reject the paper. Please take his comments seriously when preparing a revised version. This will increase the likelihood for final acceptability.

Reviewer 1 ·

Basic reporting

This study seems like a technique report, rather than a scientific paper because the study needs is very weak.
Insufficient introduction and background descriptions.
Poorer article structure in Discussions.

Experimental design

The research question and rationale did not well defined.
“…new sample of 29 asymptomatic adults participated in the study…” this is an inappropriate description. Insufficient demographic data was showed in this study.

Validity of the findings

The rationale and impacts of this study do not state clearly.
The authors concluded that the “…the advantage of capturing the joint ROM digitally, making them a powerful tool to register the progress of treatment or implement physical therapies based on biofeedback (e.g. rehabilitation videogames)…” for people with disabilities. It is not appropriate because this study did not provide this information and techniques in this manuscript.

Additional comments

This study seems like a technique report, rather than a scientific paper because the study needs is very weak.

The authors did not point out the key point that why they have to focus on the elbow and wrist chronic conditions and their relationship with the pain and poorer muscle function in this (the study needs and rationale do not strength enough).

This study claims the inertial sensors are low-cost and accurate digital measurements, but it is not really because several inertial sensors with 3-D high accuracy and sensitivities are pretty expensive from 100-1500 USD.

Poorer scientific writing in Abstract. The authors should objective showed the compared data in results sentences, rather than state “…These values were within the normal range when compared to literature studies…” Furthermore, this problem also found in conclusions sentences “…Particularly noteworthy is their high inter-rater reliability, often questioned in measurement tools…”. The authors should reveal scientific conclusions based on their findings.

The authors concluded that the “…the advantage of capturing the joint ROM digitally, making them a powerful tool to register the progress of treatment or implement physical therapies based on biofeedback (e.g. rehabilitation videogames)…” for people with disabilities. It is not appropriate because this study did not provide this information and techniques in this manuscript.

In the introduction, this study did not show the study rationale very clearly and too roughly.

The scientific paper should focus on the needs in clinical, many clinical and reach tools can be used to evaluate the ROM, such as motion analysis system, Fastrak system…. The authors should compare their advantages and disadvantages in this study, rather than only state the “Technological advances…”

The authors reviewed many studies and revealed large descriptions about the technical information for Inertial Measurement Units (IMUs), robotic platforms and mobile smartphones. I suggest the authors should focus on study needs.

Poorer scientific writing in the last paragraphs in the Introduction

“…new sample of 29 asymptomatic adults participated in the study…” this is an inappropriate description. Insufficient demographic data was showed in this study.

Poorer organization for the “Statistical analysis” in Methods

The explains for study findings did not enough and without appropriate comparing with previous studies in Discussions,

The conclusions should be reorganized.


English editing would be needed.

Reviewer 2 ·

Basic reporting

I think that this is a well done study describing the reliability of an IMU method in comparison to a traditional goniometer method. However I think the structure of the paper would be improved by making sure that the correct details and specifics, of interest to the reader, are stated in the research paper. Some information could actually be removed (see some examples in my comments below) to give more room for discussion and interpretations on your results.

Experimental design

I have no comments on the experimental design other then a question on why three repetitions were used? Can you motivate why this number was selected? In many reliability studies, especially when using ICC statistics, a larger number of repetitions are recommended.

Validity of the findings

On row 317-318 you state that “The lowest values for r2 were found in pronation (0.47), during the first session and in particular for examiner B, although examiner A obtained higher results.” Does this mean that elbow pronation-supination ROM more variable/less reliable than elbow flexion-extension ROM for example? Given how the different motion directions are measured, it is not surprising if flexion-extension turns out to be more reliable and easier to measure. I think this should be reflected on and discussed in comparison to other studies analyzing elbow ROM.

In row 338-340 you state that the Bland-Altmann plots show good agreement between the methods. The number on the y-axis are however too small to judge whether this is true or not. Please revise the figures 7-8 so that the numbers can be read.

Row 379 – The references given here are about ten years old. More recent studies have been done examining upper body limbs using inertial sensors. A few examples: “Monitoring of upper-limb movements through inertial sensors – Preliminary results” (2019), "Motion capture sensing techniques used in human upper limb motion: a review" (2019)."A new way of assessing arm function in activity using kinematic Exposure Variation Analysis and portable inertial sensors--A validity study (2016).

Additional comments

- The abstract do not state which movement directions you evaluate (elbow flexion-extension, supination-pronation, and so on). Please add this information.

- The introduction is very general, starting with describing all musculoskeletal disorders (row 58-64), and all different ways that IMUs can be used for (row 97-123). Please instead focus only on musculoskeletal disorders that impair the upper limb and specifically elbow and wrist movement which is the scope of your study (e.g. row 66-76). Also, the section about IMUs (row 97-123) should be shortened to focus on IMUs and upper limb. Perhaps a short section in the discussion about how this method can be generalized to other disorders in other parts of the body can be added if you think this is necessary.

- Row 143-150 should be removed (all research studies are structured in the way you describe). Instead the aim of the study (3ow 136-141) should be the final word of the introduction.

- Please also clarify and evaluate the aim further: We aimed to validate… by assessing criterion-related validity, intra-rater reliability and inter-rater reliability.

- Row 183-185: I don’t think it is necessary to state that each person is given a code number; this is common practice in order to fulfill demands about anonymity etc.

- Row 188- “Measures were taken by a universal goniometer” – please extend this description and add the name of the goniometer here instead (row 199-200).

- Row 250: I think you should add a short desricption about how “ROM end-feel” is defined, since not all readers are physiotherapists (e.g. that end-feel means that the joint has full ROM and the range is stopped by the anatomy). Also, add that ROM stands for Range of Motion the first time you use this abbreviation.

- Row 279 – Please do not specify which options you selected in SPSS, this may change over time due to software updates and is unnecessary information. Also, some information about the statistical choices you have made is more appropriate for a text book than a research paper even if the information is correct, e.g. row 283 “The use of SEM in reliability studies is justified because when a test administered to 284 one individual is repeated a number of times the observed value will vary between repetitions 285 due to the error in the measuring process. These responses are related to the mean. It can be proven that the true value of the measure falls within the observed value ± SEM 68% of the 287 times (McKenna, Cunningham & Straker, 2004).” Perhaps this is enough information to the reader of this paper: “The SEM and the MDC at the 90% confidence level were calculated for both measuring instruments. SEM can be estimated from the ICC ….”

- Row 334-336 – is something missing in this sentence? “as it was expected”? What was expected?

I did not find any raw data, i.e. an excel file with ROM measurements or similar? Have you considered to add such a file?

---

## Round 0.2 · accepted · Accept

Thank you for having taken into account the reviewer‘s comments and for having taken seriously their concerns.

Reviewer 1 ·

Basic reporting

The authors have reorganized added more exhaustive literature review in the introduction and discussion sections, well structure paragraphs, and sufficient context was provided to show the objectives and relevance in this study.

Experimental design

Research questions well defined, relevant, and meaningful.
Method sections described with sufficient detail.

Validity of the findings

Conclusions are well stated and strongly linked to the original research question.

Additional comments

I suggest this article is suitable and ready to publish.